# Siberian Animal Style: Stylistic Features as Generic Indication

**Elena Fiodorovna Korolkova**

The State Hermitage Museum, 191181 Petersburg, Russia; oaves@hermitage.ru

**Abstract:** This paper is devoted to the problems of differentiation of stylistic variants in the common phenomenon of the so-called Scythian and Siberian animal styles, which is one of the main distinctive features of Eurasian nomadic art. The animal style is a concept of more scale than an artistic style proper which distinguishes with some formal characteristics and depends directly on generic traditions and ethnic and cultural roots of art. Together with the technical-technological methods these formal features could be evidential indications of the origin of works of art. The Siberian collection of Peter the Great includes some different groups of golden ornaments decorated in animal styles of different origins. The paper focuses on a compact group of items originating from various mostly unknown sites from different territories in Asia including the Oxus treasure, several items from the Siberian collection of Peter the Great from Southern Siberia, a few jewelry pieces from other collections of the world museums as well as items made of leather and felt coming from the First and the Second Pazyryk kurgans. A distinctive feature of this group of zoomorphic images are colored inlays that accentuate a hind-leg or a shoulder of the animal; such inlays have the form of an intricate figure made up of a circle and a curvilinear triangle abutting to it or elongated round brackets. Genetically, such an ornamental motif, which is not generally typical for Persian art, may be linked to a periphery area of the Iranian world and nomadic culture, while the group of sites can be dated back to the 4th–3rd centuries BC. The paper considers a bracelet from the Siberian collection of Peter the Great which is the only item in this category of jewelry type of bracelets. It represents a rare type of ornament with a multi-component structure. It consists of three open-work strips with zoomorphic compositions in an animal style similar to the above-mentioned stylistic group. All three parts of the bracelet are created in a unified style, but obviously in different individual manners. There is no doubt, that the zoomorphic images show three different authors' hands, and were made by different artisans. So, there is evidence of collective work on the object when each artisan makes his own operation to create a unique jewel at a workshop. Some parts of the composition on the bracelet are similar in style to zoomorphic images from kurgan Issyk in Kazakhstan which perhaps were made in the same workshop. This fact confirms the assumption of the origin of some of Siberian jewelry.

**Keywords:** Siberia; animal style; nomadic art; Peter the Great

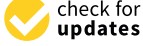



A cultural and artistic phenomenon of the so-called Scytho-Siberian Animal style is one of the main distinctive features of Eurasian nomadic art. At the same time, Animal Style art is a very complicated phenomenon with numerous local variants and it could not be determined as something homogeneous. The nomadic world is various, movable, and receptive, and there are a lot of traits of different contacts and impacts reflected in nomadic art. Nevertheless, none of the cultures with which Eurasian nomadic tribes came into contact possessed anything comparable with so-called Scythian Animal Style art. Nomads adopted some features, zoomorphic motifs, and animal images, which were consonant with their own ideological and mythological conception. Relations with the peoples occupying different parts of the Eurasian steppe zone remained for a long time a source of the exchange of artistic ideas and elements, which formed various versions of the Animal Style. Different peoples certainly practiced a kindred art with a common

mythological and ideological base, but nomadic art from different regions is characterized by different formal features.

Style in ancient art is a category, which is similar to the phenomenon of type concerning the sphere of material culture in archaeological studies (Klein 2016, p. 69). We should try to trace the cultural impulses in style and technique indications in the Animal Style of Scythian period among the Siberian art materials including the objects from Altai burial mounds. This is a complicated task since the cultural links could have been indirect.

There is a problem with where these objects came from and where they were produced. Another problem is who made them. Archaeological materials document the exchange of goods, technology, and people from across the Eurasian steppe (Linduff 2006, pp. 358–70) to and from nomadic tribes and settled oriental civilizations of highly developed culture and industry. Questions exist regarding how the artistic style could be formed, how it changed, and how it developed, enriching with different impacts.

Southern Siberia is a region neighbouring great ancient states, such as Achaemenid, Iran, and China. The great Scythian zone was traversed from west to east but was also crossed by nomadic passes from south to north. Southern Siberia lay in the territory of the intersection of the ways of mobile nomadic peoples who were very susceptive to outer cultural influences, especially in Animal Style. Eurasian nomadic culture shows evident signs of different influences from other cultures, which belonged to settled societies including Chinese, Iranian, and Greek societies. We find sometimes mixed features of Persian and Chinese stylistic traits in the same objects. Additionally, there is not any contradiction in fact of some formal adoptions from different sources and borrowing of some elements. It was a creative and natural process, which was stimulated by some ethno-political situations.

For instance, a possible historical explanation of the penetration of some Iranian impacts is a Yuezhi nomadic migration from Central Asia to Altai after Alexander the Great's conquest and their return to Bactria around 130 BC (Francfort 2020, p. 135). Thanks to archaeological investigations we can trace the great cultural changes that took place across the wide area inhabited by nomadic tribes following the great wars and other social upheavals. In particular, Alexander the Great's campaigns have been a catalyst for significant change in different areas. The Asian world has got a great impulse for movement and change. We can therefore assume that it was due to these circumstances that such a characteristic style of jewelry was found in the graves of nomadic noblemen from the 4th–3rd centuries.

Sometimes the signs of different sorts of contacts reflect in the objects of mixed character with heterogeneous parts joined, sometimes new hybrid forms arose under the influences. For example, I refer to a mirror from Grave 2 in kurgan1 of Filippovka I burial ground (Figure 1) (Okorokov and Perevodchikova 2020, p. 41, Figure 7:1). Stylistic features of the piece of art, as well as in some other objects from this grave show mixed nature and there is no doubt that the things were made of heterogeneous details. These details show features of different cultural and technological traditions. This evidence sets suppose, that the disk of a silver mirror was made in a workshop of Achaemenid Iran, but its handle belonged to nomadic tradition. So, some pieces of art had a hybrid nature and were produced by artisans of different cultural origins. This fact could be explained by the exchange of things and following using some imported details for the creation of hybrid objects. Quite possibly, jewelry workshops were located not only in the territory of the Achaemenid Empire, but also in the neighbouring periphery, and nomadic artisans could participate in the jewelry production together with goldsmiths of varied origins. Such workshops united artisans of different origins and served not only as production centers, but also as cultural centers and at the same time as professional and artistic schools. Just these centers were the points of formation and development of Animal Style.

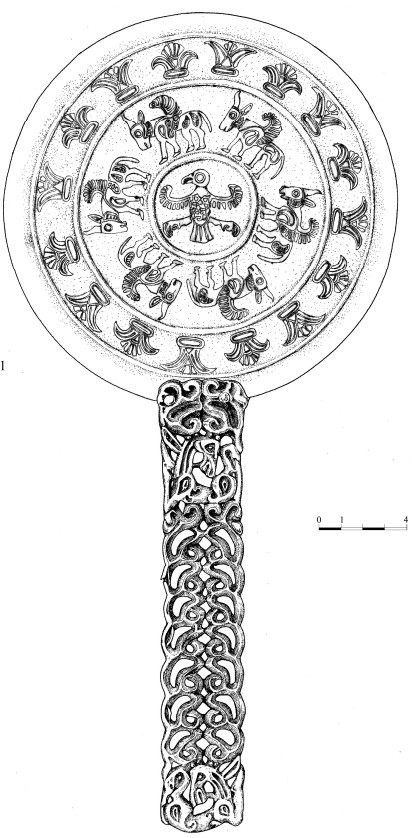

**Figure 1.** A mirror from Filippovka 1, kurgan 1, burial 2.

This is only typical of the Eurasian nomadic zone inhabited by peoples, who were in contact with both civilizations, Iranian and Chinese. The question is that of the mechanism of the mix of visual imagery and artistic forms, which occurs in objects of the same or very close type. Sometimes we find objects which consist of heterogeneous details that obviously belonged to different cultures and combined with each other evidently in the workshops somewhere available to the nomadic territory. Goldsmiths who worked for nomadic nobility were mostly from settled societies. Yet we cannot make an assumption that nomadic people who created such splendid art of Animal Style and used to make a lot of artistic things of organic materials could not produce any metalwork. They were fantastically creative and the Animal Style was born in nomadic midst and space.

Many works of art made of different materials from the Altai barrows of the Scythian period show a strong resemblance with the art of Achaemenid Persia. This fact should especially be emphasized in the case of the archaeological materials from Pazyryk kurgan 1 and kurgan 2. Some such objects are undoubtedly imported from the territory of the Achaemenid Empire. Yet there are no grounds to consider all works of applied art from Pazyryk barrows as imported goods. This art was inspired by the influence of Iranian art and we can find some very typical features borrowed from Iranian culture and involved in nomadic Animal Style. It is of great interest that at the same time and even in the same objects from Altai barrows, we can find images of griffins. These images derive from ancient Greek culture but reached the Siberian territory indirectly. They appeared in the Altai region from the 4th–3rd century BC and no earlier and were not widely spread in the Siberian territory. They occur in Pazyryk Animal Style and embellish the details of horse trappings (saddle-clothes and other things), perhaps, instead of the authentic fantastic beaked monsters being equivalent in meaning to them. The authentic Siberian monsters combine the features of different species—beasts of prey, deer, eagles, and other creatures. They are more typical to the eastern part of the Animal style area and never occur in the European Scythian art. The typical Siberian Animal style monsters similar to griffins, but never with the lion's body as in Greek mythology, could be determined

as "mythological eagle" (Rudenko 1958, pp. 101–3). Yet the Pazyryk griffins differ from the Greek and Persian ones. They have their own distinctive features which look similar to specific ornamental elements—a geometrical motif in the shape of a combination of a triangle with a circle on the hindquarter or on the shoulder of the monster's body. This ornamental element is not of Greek origin and indicates the Near East influence.

Golden ornaments from the Collection of Peter the Great are doubtless objects of great artistic merit, as is true of the things made of wood, leather, fur, and other organic materials from Altai burial mounds. Both of them sometimes show very similar decorative elements, motifs, and images in spite of making in different materials. There is no reason to suppose that all the things of Animal Style from Altai barrows were imported or produced by foreign craftsmen. Marks of these tendencies could be obviously traced especially in the art of Animal Style of the Scythian period, both in style and in technique treatment.

The Animal Style is a concept of more scale than a formal artistic style, and each local stylistic variant is distinguished by some formal characteristics. These characteristics depend directly on generic traditions and ethnic and cultural roots of art. Together with the technical-technological methods, these formal features could be evidential indications of the origin of works of art.

The Siberian Collection of Peter the Great is the first Russian archaeological collection. It contains about 250 extraordinary golden pieces in Animal Style, which served as specific ornaments in Eurasian nomadic cultures in Siberia in ancient times from the 7th century BC till the Middle Ages. All these objects were discovered in the early 18th century in course of Russian pioneers and cartographic expeditions intended to investigate unknown territories in Asia on purpose to lay the new trade ways to connect West and East. Excavation of ancient gold could be considered as a concomitant. On their way, they also searched for sources of gold, natural deposits, or ancient burial mounds rich with gold objects. These teams of pioneers excavated a lot of ancient burial mounds and dug out numerous gold objects, some of which were later were accumulated in Peter the Great's Siberian collection. That is why all the information about the exact location and territorial position of the excavated kurgans was, unfortunately, lost.

The collection was gathered, mostly owing to the efforts of Prince Matwey Gagarin, the military governor of Siberia, who executed the direct tsar's orders concerning the ancient gold from Siberian burial mounds excavated by diggers. This collection comprises really unique but heterogeneous gold objects from unfortunately unknown archaeological sites. These items are from various cultural sources. The provenance and attribution of the gold ornaments, which came into the Siberian Collection of Peter the Great are still much debated. The lack of an archaeological context for any of them, however, has so far hampered any decisive conclusions. The objects from the Peter the Great treasure were found somewhere in the vast territory of Siberia. The only way to determine the origin of each item is to compare the stylistic features and manufacture traces which could help these objects to be attributed and dated.

The main difficulty of dealing with the objects from Peter the Great treasure is unknown provenance which makes it impossible to specify the cultural center or nomadic group in which these gold ornaments were produced. The gold was dug out from the graves located somewhere within the area from modern Kazakhstan to the Altai mountains. The various personal ornaments among the finds may have come from the region between the Irtysh and Ob' rivers and even from the region of the Syr-Darya and Amudarya basins. Additionally, we can suppose that besides the Iranian cultural centers with highly developed goldsmith workshops, there were some local workshops in southern Siberia, probably in the Altai region, where some of the belt plaques in animal style could be cast or hand-made from the sheet.

At the same time, the jewelry from the Peter the Great Collection shows some common features in artistic images, which does not afford ground for taking these objects under consideration as separate cultural phenomena united owing to import and the exchange of goods only. The interaction of nomadic and sedentary societies is a major feature and,

perhaps, a main problem of human history at all times. Nomads adopted certain features of sedentary culture, transformed these according to their own necessity, and being mobile, were able to transfer these far beyond their point of origin.

Belonging to culturally similar nomadic groups, all these items demonstrate a lot of different variants and tendencies in artistic and technological aspects, according to ethnical and cultural appertaining. Marks of these tendencies could be obviously traced especially in the art of Animal Style of Scythian period, both in artistic style and in technique.

It should be noted that the colourful applied art of the Altai nomads of Scytihan times which is represented with the objects of different organic materials is extremely consonant in aesthetic aspects with Achaemenid Persian polychrome jewelry. The colouration of textiles, felt, and leather Siberian pieces is traditional to this culture as well as polychrome jewelry. Some of the types of things and techniques could be borrowed by nomadic tribes from different cultural sources, but only in the case of accordance with their aesthetic taste and conformity with the potential of their own traditional meaning.

Peter the Great's collection includes some different stylistic groups of golden ornaments decorated in Animal Styles of different origins. Some of them are derived from the Iranian world. Other ones are closely related to Chinese influences. A few things show the common features, distinguishing them from the others.

For instance, there is a compact stylistic group of items discovered in different territories in Asia, which are marked with similar decorative elements. It looks similar to colored inlays that accentuate the hind leg or shoulder of the animal; such inlays have the form of an intricate figure made up of a circle combined with a curvilinear triangle abutting it or with elongated round brackets. Genetically, such an ornamental motif, which is not typical for Persian art, may be linked to a periphery area of the Iranian world and nomadic culture, while the group of monuments can be dated back to the 4th–3rd centuries BC. This decorative element is derived from very early times and shows a strong resemblance with Assyrian art.

This group can be compared with similar objects from other world museums including the Oxus treasure from the British Museum and others. Indeed, several items from the Siberian collection of Peter the Great from Southern Siberia, this stylistic group comprises a few jewelry pieces from other collections as well as items made of leather and felt coming from Pazyryk 1 and Pazyryk 2 kurgans (Korolkova 2017, pp. 50–60; 2020, p. 220). In the Siberian Collection, we can single out a group of ornaments with zoomorphic images which show a strong resemblance with similar objects from other collections. This group comprises some ornaments of various functions: two collars, a bracelet, a roundel, a pair of belt-buckles with an animal combat scene showing a winged monster attacking a recumbent horse (Figure 2), and a splendid ornament with a fantastic bird of prey attacking a wild goat (Figure 3). All these things are embellished with inlays embedding the stones in cavities with the outlined cells, composed in a geometrical pattern with a circle in the center, which is flanked with one or two curved triangles. Such an ornamental composition usually marks the shoulder and the hindquarter of an animal figure. The same ornamental pattern we can find on the leather and felt goods from Pazyryk burial mounds, which are generally regarded as having been made under the great influence of the Achaemenid culture. Most of them were discovered in the Pazyryk 1 and Payryk 2 kurgans.

It should be noted that we never place the above-mentioned geometrical decorative element in Persian art proper, in spite of the artistic tradition in Achaemenid art to mark an animal's shoulder and hind quarter with a special "drop"-shaped accent.

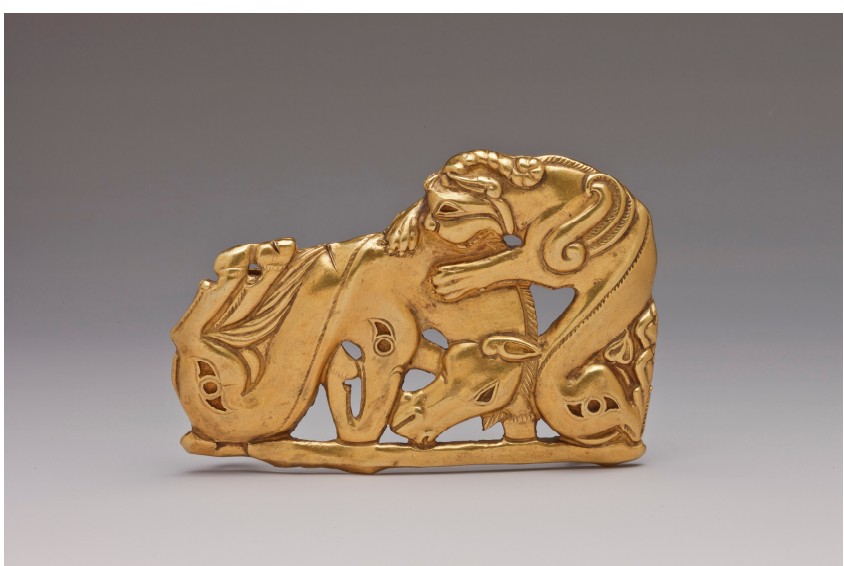

**Figure 2.** A belt-plaque from the Siberian Collection of Peter the Great. State Hermitage. Inv. Si 1727 1/5.

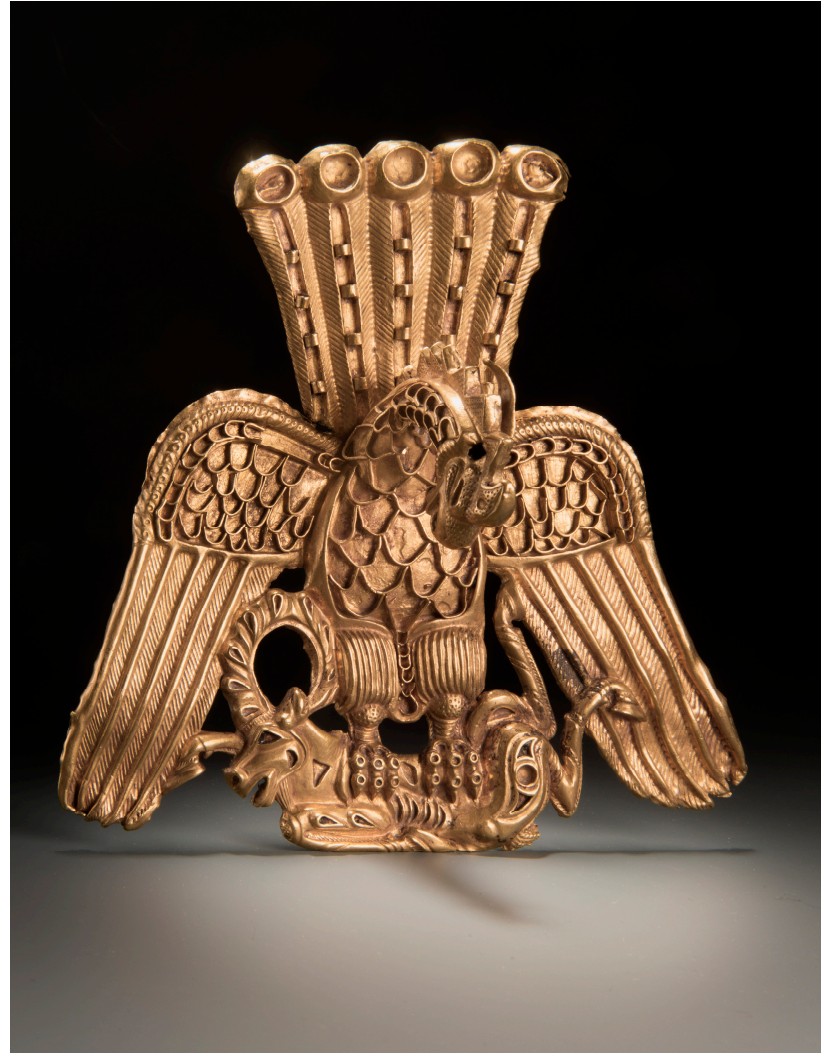

**Figure 3.** An aigrette from the Siberian Collection of Peter the Great. State Hermitage. Inv. Si 1727 1/131.

Some of the objects from the Siberian Collection bear the clear imprint of Achaemenid Persian art, and they are marked with similarly specific ornamental motifs. The most splendid and typical one is a gold torc with open ends terminating in lion griffins (Figure 4). This type of monster is similar in style to the Iranian images and similar to the murals of Susa. The treatment of the cloisonné technique is exactly like that of a gold armlet from the Oxus Treasure. This fact was doubtless noticed by O. M. Dalton (Dalton 1964, pp. 52–53). He concluded that the two ornaments should have been considered synchronous and closely related. He supposed that the torc travelled north from the Persian border, perhaps from Bactria, or that a craftsman from the south, migrating to Siberia, there carried on his tradition. Polychrome incrustation executed in technique cloisonné which is typical of Persian art should be considered a sign of the Iranian origin of the object or of the craftsman who has created the piece of jewelry, or, at least, of an artistic tradition migration. Yet the question is of the provenance of the objects. Indeed, it becomes even more complicated.

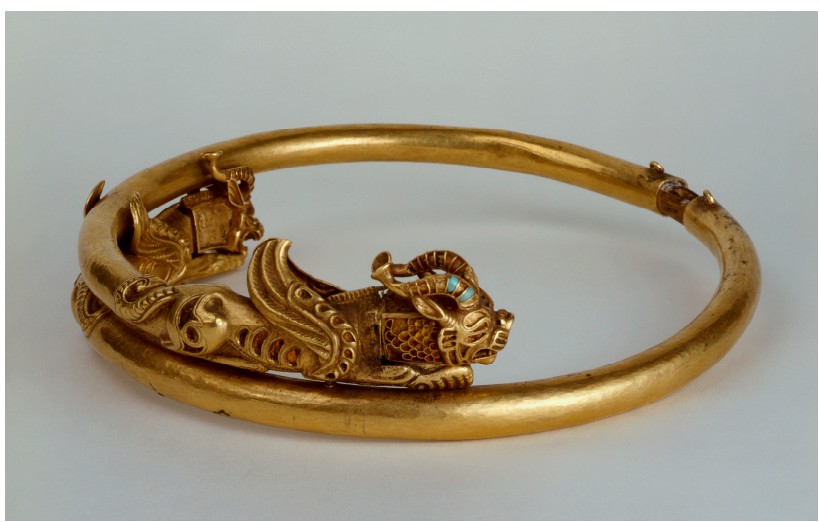

**Figure 4.** A torc from the Siberian Collection of Peter the Great. State Hermitage. Inv. Z-568.

It cannot be denied that images of monsters are also typical in the Iranian culture of Achaemenid Persia. Yet we should keep in mind that the civilization of the Ancient Near East had many various sources, which caused the complicated history of this region. Furthermore, we should keep in mind that the artistic culture of Achaemenid Iran, in turn, imbibed different traditions of Near East civilization including Assyrian culture. There is an evident similarity in composition between the monster's figure with a curved body in the relief from the Temple of Ninutra at Nimrud of Assurnasirpal (883–859) (British Museum, Korolkova 2020, p. 221, Figure 6) and the fantastic winged and horned beast of prey in the golden belt plaques from the Siberian Collection in the scene of tormenting a recumbent horse (Figure 2).

Referring to Dalton, when the Persian Empire was established, its art was borrowed from earlier sources (Dalton 1964, p. 42). The Persian objects from the Oxus treasure were under the influence of the great imperial system. Mesopotamian mythological fancy appears in the images of griffins and other monsters in Persian art and further in Scythian culture.

Polychrome incrustation in jewelry is an evident feature of the Achaemenid culture. Yet it derives from earlier epochs and cultures of the ancient East. There is a reason to believe that the technique commonly described as orfèvrerie cloisonné, originated in the Nearer East. Initially, it was very typical to Egyptian and Phoenician applied art as well as to Assyrian art of the 9th–8th centuries BC, a culture also rich in cloisonné technique. Its great development took place in Achaemenid Persia. Then from Assyria and Persia, this type of art entered the Scythian world and the region of the Oxus treasure. The monsters of the Iranian world look similar to winged creatures with goat horns and beaked eagle

heads, or lion-headed hybrid animals with lion or ungulate bodies. Such hybrid figures decorate the ends of the Hermitage torc (Figure 4. State Hermitage, Z-568). The closest parallel to this torc in style and technique is a pair of bracelets from the Oxus treasure (BM ANE 124017, Victoria and Albert Museum, 442–1884). These jewelry pieces belonged to the Iranian workshop of a "royal circle", in spite of the opinion of some who derive this collar from the Siberian art influenced by Achaemenid jewelry (Rudenko 1962, pp. 18–19). Yet, different scholars had different views on the provenance of this torc and derived it from Eastern Iran, as well as the next object (Ivanov et al. 1984, cat. 2, 3). Another golden spiral torc from the Siberian Collection (Figure 5, State Hermitage, Si 1727 1/62), with open ends terminating in feline beasts of prey (perhaps, tigers) demonstrates a very similar style and similar decoration technique, with inlays but without cloisonné. This gold torc from the Peter the Great Collection is probably dated from the 4th–3rd century B.C.

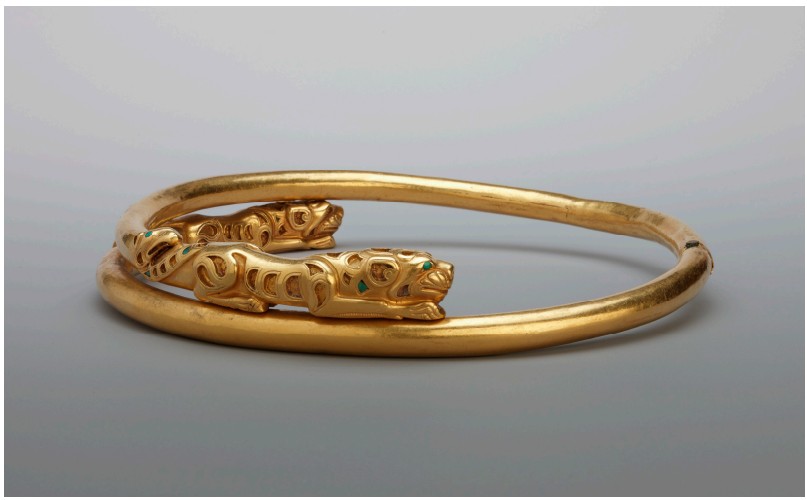

**Figure 5.** A torc from the Siberian Collection of Peter the Great. State Hermitage. Inv. Si 1727 1/62.

The figured terminals of the torc were embellished with turquoise inlays. The feline tails end with griffin heads, while ears, ribs, and other body surfaces are set with inlay cells for turquoise. The style of this object is comparable with the Pazyryk style as well as with the pieces from the Oxus treasure and from Issyk barrow (Kazakhstsan). Scholars defined this torc as an example of native Siberian goldwork influenced by Achaemenid art (Rudenko 1962, p. 18; Artamonov 1973, p. 169; Farkas 1973, p. 83). Probably, there is a reason for this conclusion. At the same time, it should be noted that we cannot consider the two above-mentioned collars as separate examples of ancient jewelry. Their relation is undoubted since they show a strong resemblance in goldsmith skill in spite of the distinction of decoration. The torc with the feline-shaped terminals could be considered an intermediate variant between the "royal school" and the peripheral cultural circle. It should be taken as evidence of a strong Iranian influence on jewelry arts in the vast territory of the Achaemenid empire and neighbouring area. This intermediate link closely connects with nomadic Pazyryk culture.

Some scholars (Artamonov 1973, p. 168; Farkas 1973, p. 83) supposed that the terminals of this torc were made in cast technique, but in spite of this opinion, it should be noticed that now we can contend well-grounded that the feline figures were hand-made on matrix from two parts (Mongiatti and Korolkova 2020, pp. 327–54).

As to a splendid gold ornament with a fantastic bird of prey attacking a wild goat from the Peter the Great collection (Figure 3; State Hermitage, Si 1727 1/131), most scholars considered this object of unknown function to be an aigrette. The ornament represents a fantastic bird of prey with its wings spread, holding a wild goat in its talons. This object combines the two methods of polychrome decoration and shows both cloisonné and inlays embedded in cavities on the shoulder and on the haunch of the goat.

The Altai parallels show some similar compositions in the decorative elements of funeral horse trappings. It is reasonable to suppose that this scene is of great semantic significance. Yet, the object may have served rather as a chieftain's aigrette than as a frontlet of a horse's headgear. The aigrette from the Siberian Collection is a rare object which shows a strong resemblance in style and ornamental details both to the jewelry from the Oxus treasure and to the Pazyryk male headgear adornment recently reconstructed by Elena Stepanova (Stepanova 2017, p. 113). Recent investigations and restoration work dealing with felt and leather details of a piece of male headgear from the kurgan Pazyryk 2 at the Hermitage museum was revealed to be very similar to the golden aigrette from the Siberian collection of Peter the Great in composition and Animal Style. According to the Altai parallel, it is reasonable to assume that this ornament may have been used as an adornment of a sacred male headgear. Compared to the Oxus treasure bracelets and aigrette, the workmanship of the Siberian aigrette is somewhat coarse. We should assume the existence of another workshop of the same time and of very close cultural traditions. Yet all these pieces are similar enough to have formed one stylistic group.

Aside from cloisonné on the neck of the monster on the torc from the Peter the Great collection, a craftsman used another method of inlay in the same object, in the cells filled in with turquoise. Such a technique, in Dalton's opinion, employed a simple method of embedding the stones in cavities made on the metal sheet. The Sarmatians further continued to use both methods (Farkas 1973, pp. 80–83).

One question of great interest is the identity of the real species of prey in the monster's talons. This is important from the point of view of the definition of the origin of the object since it looks similar to Capra sibirica, which is closely related to the European Capra ibex, but has only inhabited some Central Asiatic regions including Tajikistan, Kazakhstan, Afghanistan, the Pamirs, the Tien Shan, and Southern Siberia. Thus, this object and the rest in the group seem to have been produced somewhere in this territory.

The iconography of this item with a griffin-like monstrous bird attacking an ibex is also paralleled on a felt saddle cloth from the horse equipment from Altai in Pazyryk, kurgan 1. Like other parallels from Altai in the first millennium BC, this reflects to varying degrees the interactions of Achaemenid Persia with the nomadic cultures of Central Asia, Southern Siberia, and Iran.

There are some specific features in the treatment of this object which are of great interest. For example, the spread tail of the monstrous bird is treated with tiny little tubular loops disposed of in vertical rows in the riffles between feathers. These cylinders were intended, perhaps for attachment of the missing incrustation, in the five grooves on the bird's tail feathering. It is a very strange thing, but the only technique reminiscent of such a method of embellishment occurs in ivories from Nimrud. They were discovered at Fort Shalmaneser and could be dated from the 8th century BC (Mallowan 1966, pp. 566–69, figure. 513–15). Some ivory panels, plaques, and pyxis from Nimrud demonstrate a very similar method of embellishment, despite the difference in material and chronological distance. The resemblance is so close that there is no doubt of the genetic relations between these two examples of decorative art. So, we can derive the method of decoration of the Siberian ornament from Assyrian art as well as Achaemenid Persia, where the cloisonné technique was adopted.

On the ivory panel from the Fort Shalmaneser (Iraq Museum, Bagdad ND 9475) a single piece of green glass inlay remains in the stalk of one of the papyrus flowers (Mallowan 1966, p. 568).

There is a similar example of the same type of incrustation on the small eagle figure from the Siberian Collection (State Hermitage, Z-557; Ivanov et al. 1984, p. 20, cat. 14; Korolkova 2020, p. 224, Figure 13), the neck and breast of which are decorated with cloisonné. This item was incrusted with turquoise and the eagle's tail was framed by the small cylinders to carry the inlays which are missing.

The aigrette combines the three methods of polychrome decoration and shows both cloisonné and inlays embedded in cavities accented with the shoulder and the hindquarter

of a goat, and perhaps, the above-mentioned way of incrustation with small cylindrical pieces.

Neither Scythian nor Sarmatian periods in Animal Style art demonstrate the use of lapis-lasuli to incrust the ornaments preferring to inlay turquoise or blue glass-frit mass, as distinct from the Persian culture. This tradition is rather a distinction of ancient Central Asiatic cultures. To clarify the technique, it should be stressed that the inlays on the torc were embedded in cavities not cut, but hammered on the matrix. Aside from some common features, including the above-mentioned decorative element in the shape of a triangle combined with a circle, it should be noted that another specific feature of the embedding of inlays is a double-line contour that outlines the cell of inlay. This feature is one of the indicators of the close in style and cultural origin group of jewelry.

Among the items of this stylistic group are a couple of the belt plaques from the Siberian collection of Peter the Great, with an animal combat scene showing a winged monster attacking a recumbent horse with reversed hindquarters (Figure 2, State Hermitage, Si 1727 1/5, 6). The objects of this stylistic group are mostly made via methods of cold treatment of metal. These belt plaques are also chiselled, which is usual in this cultural tradition.

Recurring to the subject of the specific group of ornaments with inlays, we should mention some additional analogy items which are little known.

One of them is a bracelet from a private collection. The Animal Style design adorns the bracelet and represents a fantastic image with mixed features of different animals. This item, no doubt, belongs to the same group of ornaments as the above-mentioned objects from Peter the Great's Collection (Korolkova 2020, p. 226, Figure 15).

Another one is presumably the hilt of a dagger with a double feline figured composition comparable with some Persian examples as well as very late items of Indian weapons. This hilt is unfortunately lost and it is represented only in the drawing of the collection of I. G. Messershmidt, a German scientist of the early 18th century who was sent to Siberia to investigate this region and its history by Peter the Great. The drawings of the lost objects from his collection are preserved in the archives of the Academy of Science in Saint Petersburg (Korolkova 2020, p. 226, Figure 16).

All these objects are marked with the imprint of influences of Near East civilization and continue the development of cultural traditions which derive from Assyrian and Persian cultures. We still do not know the center of their production, but it evidently belongs to the territory bordering Achaemenid Iran and the nomadic zone and, perhaps settlements in Central Asia.

The question of Assyrian, Iranian, and notably Greek and Chinese influences in Scythian Animal Style art was under consideration by Gregory Borovka, who has emphasized that "it has assimilated and adapted foreign motives without impairing the vigour Scythian craftsmanship or its devotion to traditional subjects and conventions" (Borovka 1928, p. 6). However, G. Borovka (Borovka 1928, pp. 88–89) supposed that the Altai region was not the cradle of the original Animal Style, but instead the great centre of its expansion when it had attained maturity. He stressed the enormous importance of connections subsisting between the Scythian world and ancient China in the process of forming the Animal Style art (Borovka 1928, p. 7).

There is another golden object that could be included in this stylistic group. This is a wide bracelet from the Siberian collection of Peter the Great (Figures 6 and 7. State Hermitage, Si 1727 1/68), which is the only item of this category of jewelry in the collection of this type of bracelet. It represents a rare type of armlet with a multi-component structure. It consists of three open-work friezes with zoomorphic compositions in an animal style similar to the above-mentioned stylistic group, related to Iranian culture. All three parts of the bracelet are created in a common style, but obviously in different individual manners. There is no doubt that the zoomorphic images show three different authors' hands, and that the parts of the bracelet were made by different goldsmiths. Thus there is evidence of collective work on the object, during which each artisan creates their own method of

creating a unique jewel at a workshop. This could be explained, for example, by a lack of time, as in the case of working on items for a funeral.

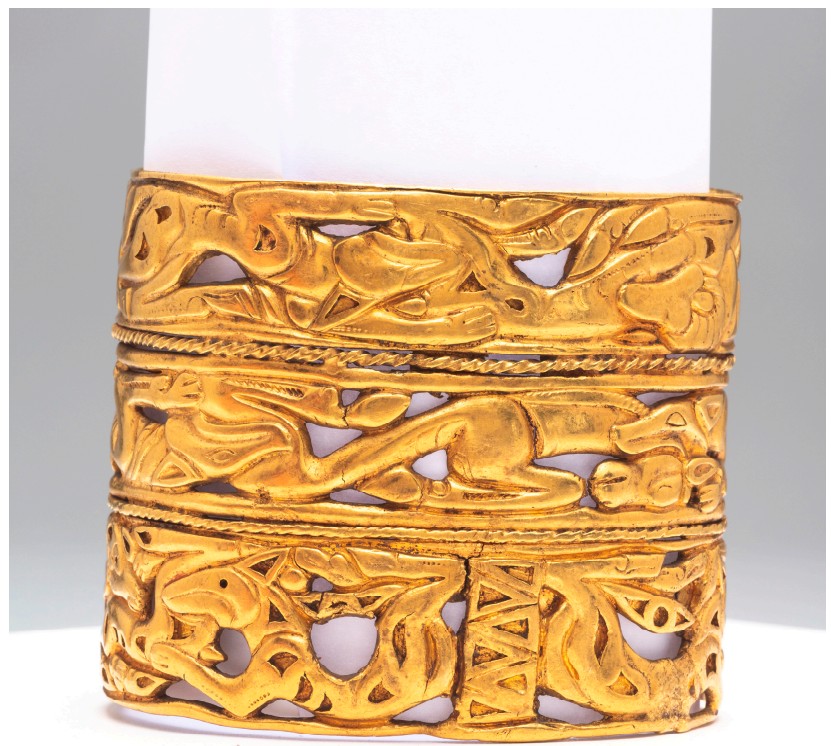

**Figure 6.** A bracelet from the Siberian Collection of Peter the Great. State Hermitage. Inv. Si 1727 1/68.

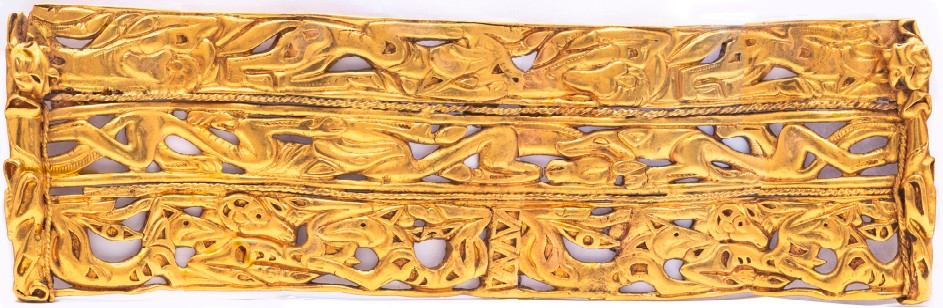

**Figure 7.** A bracelet from the Siberian Collection of Peter the Great. State Hermitage. Inv. Si 1727 1/68. Panorama view.

In each of the three friezes a typical iconic pattern occurs, such as reversed hindquarters of animal bodies. Some parts of the composition on the bracelet are similar in style to zoomorphic images from the kurgan Issyk in Kazakhstan which were perhaps made in the same workshop (Figure 8). This fact confirms the assumption of the origin of some Siberian jewelry. The central frieze of the bracelet shows great artistic merit, and professional methods of a clear and balanced composition, which are connected with Central Asian traditions. No inlays adorn this golden strip.

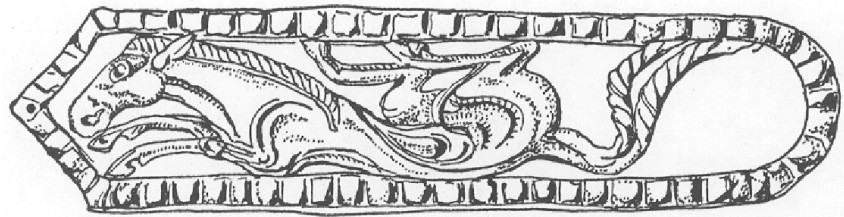

**Figure 8.** An ornament from a dagger. Issyk kurgan. Kazakhstan.

One of the outer friezes was made by a goldsmith, who represented the artistic tradition similar to the above-mentioned group of things with inlays of specific forms, which derives from Achaemenid Iranian culture. All three friezes show a strong distinction in numerous iconic features, such as proportions of animal figures, and treatment of details, which prove the work of three different persons. More than that, the three elements of the bracelet show a great difference in the professional and artistic level of the goldsmiths.

The third frieze was executed by a craftsman of less artistic merit, who tried to follow the ways of decoration and composition adopted from the Iranian tradition. All the strips were made with a technic of casting with a lost vax model. Participation of three different persons of different cultural origins in the process of manufacturing one object, as well as their collective work in a goldsmith workshop, should be considered a very important observation, which shdes light on the matter of artistic and manufacturing features of the ancient world and the development of nomadic Animal Style art.

A similar situation was observed and noted by Konstantin Chugunov in the case of a study of a dagger with decoration in Animal Style from the kurgan Arzhan 2 in Tuva, where two different styles and technical methods of zoomorphic decoration were obviously determined in one object. This fact was explained by the author of the article as an example of the cooperation of craftsmen in jewelry production of different origins and different traditions (Chugunov 2004, p. 73). Traces of different cultural and artistic influences can be observed in jewelry style and technique. A similar situation was alluded to by Vladimir Kisel concerning the early Scythian art of the Caucasian region and the Black Sea area (Kisel 2003, p. 135). Such a situation seems to be typical of the nomadic world. The main issue is establishing how and why such jewelry items arrived in Siberia and who produced them. Yet the question of the provenance is more complicated.

Archaeological research has revealed the extent of the exchange of goods, technology, and people across the Eurasian steppe between nomadic tribes as well as sedentary Oriental civilizations that were highly developed both culturally and technologically.

The so-called Scythian "Animal Style" shows the presence of different stylistic forms and artistic traditions, but its meaning, content, and imagery remained quite uniform throughout a vast territory. The region of Southern Siberia can be considered the focal point for ancient cultural interactions between different Asiatic peoples and nomadic tribes in the first millennium BC.

We have to recognize at least two vectors of cultural and artistic influences on Animal Style art, from northern China and from the Iranian world. The only region where a conjunction of mixed tendencies of stylistic features from the Iranian world, Central Asian, and northern Chinese traditions as well as gold-working techniques could be observed in the same culture as the organic artistic phenomenon connected with the ethnic group, was southern Siberia, including Altai. Additionally, the existence of the local gold-working workshops in this territory is not out of the question.

The ancient Eurasian nomadic world was at once a heterogenous conglomerate of peoples and cultures, and a very permeable environment, extremely receptive to foreign influences. The nomads borrowed visual patterns and imagery from other peoples and wove them into their own mythological context. Since the Eurasian nomads inhabited a vast region that was surrounded by several major cradles of civilization (China, Iran, Mesopotamia, Anatolia, Greece), the Scythian "Animal Style" does not represent a single

artistic tradition but had many sources and versions, in spite of being, on the whole, an artistic phenomenon characteristic exclusively of the Scythian world.

**Funding:** This research received no external funding.

**Conflicts of Interest:** The author declares no conflict of interest.

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
