# Peer review of "Siberian Animal Style: Stylistic Features as Generic Indication"

_arts, 2022_

Round 1

Reviewer 1 Report

The various archaeological cultures that form the Scythian-Siberian world are united by a special complex of things - the so-called Scythian triad. It includes weapons, horse equipment and decorative and applied art of the "animal style" - the images of which embellished mainly the same bridle and weapons. Zoomorphic art of nomads of the Scythian-Siberian community and their closest neighbors is a separate artistic direction. The Scythian-Siberian "animal style" was formed insofar as in the conditions of the economic, ethnic and linguistic community of the Scythian-Siberian world, common ideological ideas were formed in this world. The mobility of nomads helped to exchange themes, plots, visual techniques, skills of modeling works of art. In addition, there was reflected constant influence of the rich and sophisticated zoomorphic art of the Near-Asian civilizations (Assyria, Urartu, the legacy of the Hittite Kingdom, the "Luristan style" of Iran, Achaemenid Iran). The western part of the Scythian-Siberian world was more and more influenced by ancient Greek and, to a lesser extent, by Thracian zoomorphic art. And in the eastern part of the Scythian-Siberian world the influence of the art of Ancient China was rather appreciable.

The reviewed article is devoted to the aspects of the influence of Achaemenid Iran and China on the Siberian part of the Scythian-Siberian "animal style". The author demonstrates a good knowledge of specialized literature. The author’s competent stylistic and technological analysis allows him to come to reasonable conclusions about the influence of Near Eastern art (mainly of Achaemenid empire, and in one concrete case – of Assyrian empire, thus mediated by some yet unknown intermediate forms) and of Chinese art on the creation of certain items related to the Siberian Collection of Peter the Great. The objective ambiguity of the localization of objects from this collection and the problem of their narrow dating prevents the building of clear evolutionary chains, but this is a matter of the future (taking into account the ongoing accumulation of archaeological material), and the reviewed article brings closer the possibilities of clarifying the chronology and establishing the technological and stylistic origins of these analyzed masterpieces of art of the the Siberian Collection of Peter the Great.

Author Response

Thank you for your review.

Reviewer 2 Report

1. This presentation frankly lacks a clear structure, a clear and comprehensible subject and object of study, references and accompanying illustrations.  

2. The key subject of the study - triangles, circles and scrolls - should be placed in a separate table, which would also show all the analogies that the author mentions (for example, from Pazyryk). Also references should be made to the analogies when speaking about the Assyrian origins of this type of images. And in all other cases when it comes to Achaemenid and Chinese parallels. 

3. About  bracelet from Siberian collection of Peter I  and the conclusion that it was made by three different masters, with great differences in professional and artistic level.  Apart from unsubstantiated assertions, some kind of argument needs to be made. 

Author Response

 To my regret, the reviewer has not caught the main idea of my article and has taken geometrical motives (triangles, circles and scrolls) as the key subject of the study. According to this understanding, the text demands additional illustrations, tables and references to the analogies (for example, from Pazyryk). According to such opinion, references should have been made to the Assyrian images, Achaemenid and Chinese parallels.  Nevertheless, the reviewer is wrong: this subject was the key study of my previous articles published in 2017 and 2020 (See: Korolkova E. F. O proiskhozhdenii nekotorykh osobennostei sibirskogo zverinogo stilia // Kratkije soobschenija Instituta arkheologii. Vypusk 247. 2017. S. 50 – 60. (rus.)  Korolkova Elena. 'Animal Style' art: influences and traditions in the nomadic world //'Masters of the steppe: the impact of the Scythians and later nomad societies of Eurasia'.  Ed. By Pankova Svetlana and Simpson St John. Proceedings of a conference held at the British Museum 27 – 29 October 2017: Archaeopress. Oxford. 2020. P. 216 – 226), which I referred to in my present article. To my mind, there is no need to repeat again this subject, alluded to in the present text. My present article is devoted to the problems of Animal Style art and the mechanisms of development of its formal features, which sometimes depended of the outer influences and connected with technical-technological traditions. The main idea of the article is correlation of formal stylistic features with the traces of technical operations and features of technological traditions, which could be considered as generic indications.

            As for the bracelet from the Siberian collection of Peter the Great, this thing shows obviously three different artistic manners in three friezes, each with the complex of individual artistic methods in depicting of zoomorphic images. Images of the same species are so different in proportions, outlines, details and dressing in some occasions crude and brute, in others – refined and plastic, that only the person really ignorant in studies of art could not see the difference of manner.      

Reviewer 3 Report

The article is very well and logically structured. The author considers the influences on the art of nomads on the basis of known collections, the origin and chronology of which are known very approximately. Meanwhile, now there are examples of studies of objects made in the animal style, but originating from dated burials. These studies show a picture very close to the author's conclusions - the work of jewelers of different origin and artistic experience on one object. These works could be taken into account, but, probably, such a direction could be a continuation of the presented study.

Author Response

Thank you for your review. The needed changes are added.